# Age and sex-specific risks of myocarditis and pericarditis following Covid-19 messenger RNA vaccines

Stéphane Le Vu [1✉], Marion Bertrand[1], Marie-Joelle Jabagi [1], Jérémie Botton [1,2], Jérôme Drouin[1], Bérangère Baricault[1], Alain Weill [1], Rosemary Dray-Spira [1] & Mahmoud Zureik[1,3]

Cases of myocarditis and pericarditis have been reported following the receipt of Covid-19 mRNA vaccines. As vaccination campaigns are still to be extended, we aimed to provide a comprehensive assessment of the association, by vaccine and across sex and age groups. Using nationwide hospital discharge and vaccine data, we analysed all 1612 cases of myocarditis and 1613 cases of pericarditis that occurred in France in the period from May 12, 2021 to October 31, 2021. We perform matched case-control studies and find increased risks of myocarditis and pericarditis during the first week following vaccination, and particularly after the second dose, with adjusted odds ratios of myocarditis of 8.1 (95% confidence interval [CI], 6.7 to 9.9) for the BNT162b2 and 30 (95% CI, 21 to 43) for the mRNA-1273 vaccine. The largest associations are observed for myocarditis following mRNA-1273 vaccination in persons aged 18 to 24 years. Estimates of excess cases attributable to vaccination also reveal a substantial burden of both myocarditis and pericarditis across other age groups and in both males and females.

[1] EPIPHARE Scientific Interest Group in Epidemiology of Health Products, (French National Agency for the Safety of Medicines and Health Products - ANSM, French National Health Insurance - CNAM), Saint-Denis, France. [2] Faculté de Pharmacie, Université Paris-Saclay, 92296 Châtenay-Malabry, France. [3] University Paris-Saclay, UVSQ, University Paris-Sud, Inserm, Anti-infective evasion and pharmacoepidemiology, CESP Montigny le Bretonneux, France. ✉email: stephane.le-vu@ansm.sante.fr

On July 19, 2021 the European Medicines Agency advised that myocarditis and pericarditis be added to the list of adverse effects of both messenger RNA (mRNA) based vaccines (BNT162b2 [Pfizer–BioNTech] and mRNA-1273 [Moderna]) against coronavirus disease 2019 (Covid-19)[1]. This statement followed pharmacovigilance reports of an increased risk of myocarditis among recipients of mRNA vaccines that showed certain common patterns[2,3]. Several reports indicate that adverse events typically occur within a week after injection, mostly after the second dose of vaccine, cluster in young males, and result in a mild clinical course and short duration of hospitalization[4–6]. However, the predominance of a vaccine-associated risk in males[7] and its extent regarding pericarditis, as a specific condition, remains uncertain[8–11]. Population-based risks estimates for each condition and across sex and age groups and by vaccine type remains crucial as vaccination campaigns are still to be extended especially towards the youngest and with subsequent doses. The Covid-19 vaccination campaign began in France in late 2020 with the gradual roll-out of the two mRNA vaccines, BNT162b2 and mRNA-1273 alongside viral vector-based vaccines. Initially reserved for the oldest and most vulnerable groups, as well as healthcare professionals, vaccination was opened up to the entire population over the age of 18 years as of May 12, 2021, and to all over 12 years old as of June 15, 2021. As of October 31, 2021 approximately 50 million people (88% of the eligible population, i.e. over 12 years old) in France had received a full vaccination schedule[12]. Here, we aimed to estimate the age and sex-specific associations between each mRNA Covid-19 vaccine and the risk of myocarditis and pericarditis, using nationwide hospital discharge and vaccine data for France.

## Results

**Characteristics of the study population**. Between May 12, 2021 and October 31, 2021, within a population of 32 million persons aged 12 to 50 years, 21.2 million first (19.3 million second) doses of the BNT162b2 vaccine and 2.86 million first (2.58 million second) doses of the mRNA-1273 vaccine were received (Table S1). In the same period, 1612 cases of myocarditis (of which 87 [5.4%] had also a pericarditis as associated diagnosis) and 1613 cases of pericarditis (37 [2.3%] with myocarditis as associated diagnosis) were recorded in France. We matched those cases to 16,120 and 16,130 control subjects, respectively. The characteristics of the cases and their matched controls are shown in Table 1. For both myocarditis and pericarditis, key differences between cases and controls included a higher proportion among cases of a history of myocarditis or pericarditis, of history of SARS-CoV-2 infection, and receipt of an mRNA Covid-19 vaccine. The mean age and proportion of women were lower among patients with myocarditis than those with pericarditis.

**Risk of myocarditis and pericarditis associated with vaccination**. For both vaccines, the risk of myocarditis was increased in the seven days post vaccination (Table 2; in the rest of the text, we will refer to multivariable odds ratios). For the BNT162b2 vaccine, odds ratios were 1.8 (95% confidence interval [CI]: 1.3–2.5) for the first dose and 8.1 (95% CI, 6.7–9.9) for the second. The association was stronger for the mRNA-1273 vaccine with odds-ratios of 3.0 (95% CI, 1.4–6.2) for the first dose and 30 (95% CI, 21–43) for the second. The risk of pericarditis was increased in the seven days following the second dose of both vaccines, with odds ratios of 2.9 (95% CI, 2.3–3.8) for the BNT162b2 vaccine and 5.5 (95% CI, 3.3–9.0) for the mRNA-1273 vaccine. Vaccination in the previous 8 to 21 days, with either the BNT162b2 or mRNA-1273 vaccine was not associated with a risk of myocarditis or pericarditis. Independently of vaccination status, a history of

myocarditis was strongly associated with a risk of contracting myocarditis during the study period, with an odds-ratios of 160 (95% CI, 83–330). The same was true for pericarditis, with an odds ratio of 250 (95% CI, 120–540). No interaction was found between history of myocarditis or pericarditis and vaccine exposure. Infection with SARS-CoV-2 in the preceding month was also associated with a risk of myocarditis (odds ratio, 9.0 [95% CI, 6.4–13]) or pericarditis (odds ratio, 4.0 [95% CI, 2.7–5.9]).

**Subgroup estimates by sex and age classes**. The risk of myocarditis was substantially increased within the first week post vaccination in both males and females (Fig. 1 and Table S2). Odds-ratios associated with the second dose of the mRNA-1273 vaccine were consistently the highest, with values up to 44 (95% CI, 22–88) and 41 (95% CI, 12–140), respectively in males and females aged 18 to 24 years but remaining high in older age groups. Odds-ratios for the second dose of the BNT162b2 vaccine tended to decrease with age, from 18 (95% CI, 9–35) and 7.1 (95% CI, 1.5–33), respectively in males and females aged 12 to 17 years, down to 3.0 (95% CI, 1.5–5.9) and 1.9 (95% CI, 0.39–9.3), respectively in males and females aged 40 to 51 years.

An increased risk of pericarditis was also found in the first week after the second dose of either of the mRNA vaccines among both males and females (Fig. 2 and Table S3). Odds-ratios for the second dose of the BNT162b2 vaccine showed a downward trend across age groups with values up to 6.8 (95% CI, 2.3–20) and 10 (95% CI, 2.5–41), respectively in males and females aged 12 to 17 years. The second dose of the mRNA-1273 vaccine was associated with pericarditis among males and among females only within age 30 to 39 years (odds-ratio 20 [95% CI, 3.5–110]) and age 40 to 50 years (odds-ratio 13 [95% CI, 3.5–49]).

Associations between vaccination within the seven preceding days and the risk of myocarditis or pericarditis were of the same magnitude when the analysis was restricted to the period prior to the warning against myocarditis and pericarditis as adverse events sent to prescribers on July 19, 2021 (Fig. S1 and Table S4). The results were unchanged in models excluding patients with a history of SARS-CoV-2 infection in the past month, those with a history of myocarditis or pericarditis within five years, those diagnosed with both myocarditis and pericarditis, or those with a hospitalization within a month prior to index date.

**Excess events**. We estimated the number of excess cases attributable to vaccines by sex and age group (Fig. 3). The number of excess cases of myocarditis per 100,000 doses administered to adolescent males 12 to 17 years was 1.9 (95% CI, 1.4–2.6) for the second dose of the BNT162b2 vaccine and for young adults 18 to 24 years of age reached 4.7 (95% CI, 3.8–5.8) for the second dose of the BNT162b2 vaccine, and 17 (95% CI, 13–23) for the second dose of the mRNA-1273 vaccine (Fig. 3). This translates into one case of vaccine-associated myocarditis per 52,300 (95% CI, 38,200–74,100) second doses of the BNT162b2 vaccine among 12–17 years, and 21,100 (95% CI, 17,400–26,000) second doses of the BNT162b2 vaccine and 5900 (95% CI, 4400–8000) second doses of the mRNA-1273 vaccine among 18–24 years (Table S5). Estimates of excess cases were lower for older age groups and generally for females. However, the number of excess cases of myocarditis attributable to the second dose of the mRNA-1273 vaccine was consistently higher. Among females aged 18 to 24 years, the estimated number of excess cases of myocarditis per 100,000 doses reached 0.63 (95% CI, 0.34–1.1) for the second dose of the BNT162b2 vaccine (corresponding to 1 case per 159,000 [95% CI, 90,800–294,400] doses) and 5.3 (95% CI, 3.0–9.1) for the second dose of the mRNA-1273 vaccine

**Table 1 Characteristics of study cases and controls.**

| | Myocarditis | | Pericarditis | |
| --- | --- | --- | --- | --- |
| | Cases | Controls | Cases | Controls |
| | (N = 1612) | (N = 16,120) | (N = 1613) | (N = 16,130) |
| Sex | | | | |
| Male | 1281 (79.5) | 12,810 (79.5) | 989 (61.3) | 9890 (61.3) |
| Female | 331 (20.5) | 3310 (20.5) | 624 (38.7) | 6240 (38.7) |
| Age[a] | | | | |
| Mean (sd) | 27.8 (9.51) | 27.8 (9.51) | 33.4 (10.3) | 33.4 (10.3) |
| Median (range) | 25.0 (20.0–34.3) | 25.0 (20.0–34.3) | 34.0 (24.0–42.0) | 34.0 (24.0–42.0) |
| Age distribution[a] | | | | |
| 12–17 | 166 (10.3) | 1660 (10.3) | 101 (6.3) | 1010 (6.3) |
| 18–24 | 586 (36.4) | 5860 (36.4) | 312 (19.3) | 3120 (19.3) |
| 25–29 | 250 (15.5) | 2500 (15.5) | 197 (12.2) | 1970 (12.2) |
| 30–39 | 361 (22.4) | 3610 (22.4) | 465 (28.8) | 4650 (28.8) |
| 40–50 | 249 (15.4) | 2490 (15.4) | 538 (33.4) | 5380 (33.4) |
| Deprivation Index[b] | | | | |
| Most deprived | 986 (61.2) | 9567 (59.3) | 1049 (65.0) | 10,080 (62.5) |
| Least deprived | 626 (38.8) | 6553 (40.7) | 564 (35.0) | 6050 (37.5) |
| History of myocarditis or pericarditis[c] | 126 (7.8) | 9 (0.1) | 173 (10.7) | 8 (0.0) |
| History of SARS-CoV-2 infection[d] | 64 (4.0) | 107 (0.7) | 42 (2.6) | 110 (0.7) |
| Receipt of mRNA vaccine | 950 (58.9) | 7837 (48.6) | 906 (56.2) | 8436 (52.3) |

[a]At index date (date of hospital admission for myocarditis for case patients and date of selection for matched control individuals).
[b]Least deprived refers to the grouping of 1st and 2nd quintiles, and most deprived to the grouping of 3d to 5th quintiles of the deprivation index.
[c]Defined as an hospitalization with the respective condition within past 5 years.
[d]Either a positive RT-PCR or antigenic test for SARS-CoV-2, or hospitalization for COVID-19, within 30 days prior to index date.

**Table 2 Association between myocarditis and pericarditis and exposure to mRNA vaccines within 1 to 7 days and 8 to 21 days.**

| | | Myocarditis | | | | Pericarditis | | | |
| --- | --- | --- | --- | --- | --- | --- | --- | --- | --- |
| | | Cases | Controls | OR (95% CI)[a] | aOR (95% CI)[b] | Cases | Controls | OR (95% CI)[a] | aOR (95% CI)[b] |
| Unexposed | Days[c] | 1078 | 13342 | Reference | Reference | 1269 | 13398 | Reference | Reference |
| BNT162b2 | | | | | | | | | |
| Dose 1 | 1–7 | 51 | 370 | 1.7 (1.3–2.4) | 1.8 (1.3–2.5) | 43 | 398 | 1.1 (0.83–1.6) | 1.3 (0.92–1.8) |
| | 8–21 | 71 | 855 | 1.1 (0.86–1.4) | 1.2 (0.93–1.6) | 72 | 824 | 0.94 (0.73–1.2) | 0.93 (0.72–1.2) |
| Dose 2 | 1–7 | 211 | 439 | 6.9 (5.7–8.4) | 8.1 (6.7–9.9) | 93 | 374 | 2.7 (2.2–3.5) | 2.9 (2.3–3.8) |
| | 8–21 | 72 | 816 | 1.2 (0.95–1.6) | 1.3 (0.98–1.7) | 80 | 765 | 1.2 (0.91–1.5) | 1.3 (0.98–1.6) |
| mRNA-1273 | | | | | | | | | |
| Dose 1 | 1–7 | 9 | 48 | 2.4 (1.2–5) | 3 (1.4–6.2) | 8 | 78 | 1.1 (0.52–2.2) | 1.2 (0.56–2.4) |
| | 8–21 | 10 | 109 | 1.2 (0.63–2.3) | 1.1 (0.55–2.3) | 9 | 146 | 0.65 (0.33–1.3) | 0.73 (0.37–1.4) |
| Dose 2 | 1–7 | 106 | 51 | 27 (19–39) | 30 (21–43) | 26 | 54 | 5.3 (3.3–8.4) | 5.5 (3.3–9) |
| | 8–21 | 4 | 89 | 0.68 (0.25–1.9) | 0.59 (0.19–1.9) | 11 | 89 | 1.4 (0.72–2.5) | 1.5 (0.76–2.9) |
| History of myocarditis or pericarditis[d] | | | | | | | | | |
| No | | 1486 | 16111 | Reference | Reference | 1440 | 16122 | Reference | Reference |
| Yes | | 126 | 9 | 140 (71–280) | 160 (83–330) | 173 | 8 | 250 (120–520) | 250 (120–540) |
| History of SARS-CoV-2 infection[e] | | | | | | | | | |
| No | | 1548 | 16013 | Reference | Reference | 1571 | 16020 | Reference | Reference |
| Yes | | 64 | 107 | 6.3 (4.6–8.6) | 9 (6.4–13) | 42 | 110 | 3.9 (2.7–5.7) | 4 (2.7–5.9) |
| Deprivation Index[f] | | | | | | | | | |
| Most deprived | | 986 | 9567 | Reference | Reference | 1049 | 10080 | Reference | Reference |
| Least deprived | | 626 | 6553 | 0.9 (0.8–1) | 0.88 (0.77–1) | 564 | 6050 | 0.87 (0.77–0.98) | 0.87 (0.76–0.99) |

[a]Odds-ratio (95% confidence interval) were obtained from univariable conditional logistic regression, adjusting for matching variables (sex, age and department of residence).
[b]Adjusted odds-ratio (95% confidence interval) were obtained from multivariable conditional logistic regression, adjusting for all covariates and matching variables.
[c]Period of vaccine receipt relative to index date.
[d]Defined as an hospitalization with the respective condition within past 5 years.
[e]Either a positive RT-PCR or antigenic test for SARS-CoV-2, or hospitalization for COVID-19, within 30 days prior to index date.
[f]Least deprived refers to the grouping of 1st and 2nd quintiles, and most deprived to the grouping of 3d to 5th quintiles of the deprivation index.

(corresponding to 1 case per 18,700 [95% CI, 11,000–33,400] doses). The number of excess cases of pericarditis is presented in Fig. 3. As for myocarditis, estimates for the second dose of the mRNA-1273 vaccine were consistently higher.

**Characteristics of myocarditis and pericarditis cases occurring after vaccination**. Among exposed cases, the delay between administration of the vaccine and hospitalization (Fig. S2) was shorter after the second dose than after the first dose, both for myocarditis (median of 4 days versus 10 days after the BNT162b2 vaccine and of 3.5 days versus 9 days after the mRNA-1273 vaccine) and for pericarditis (median of 6 days versus 10 days after the BNT162b2 vaccine and of 3 days versus 11 days after the mRNA-1273 vaccine).

Table 3 shows the characteristics of cases acquired within 7 days of vaccination (deemed post-vaccination cases) compared to those acquired within a larger delay or in the absence of vaccination. Post-vaccination cases were significantly younger (predominantly in 18 to 24 years), more frequently concerned males for myocarditis but not for pericarditis, and without a history of myocarditis or pericarditis, respectively, or of SARS-CoV-2 infection. The lengths of hospital stay were not significantly different in post-vaccination cases of myocarditis (median 4 days) and pericarditis (median 2 days) than in unexposed cases. The frequency of admission in intensive care unit, mechanical ventilation or death was lower for post-vaccination cases than for unexposed cases. After a follow-up of 30 days after discharge, 4 (0.24%) deaths among cases of

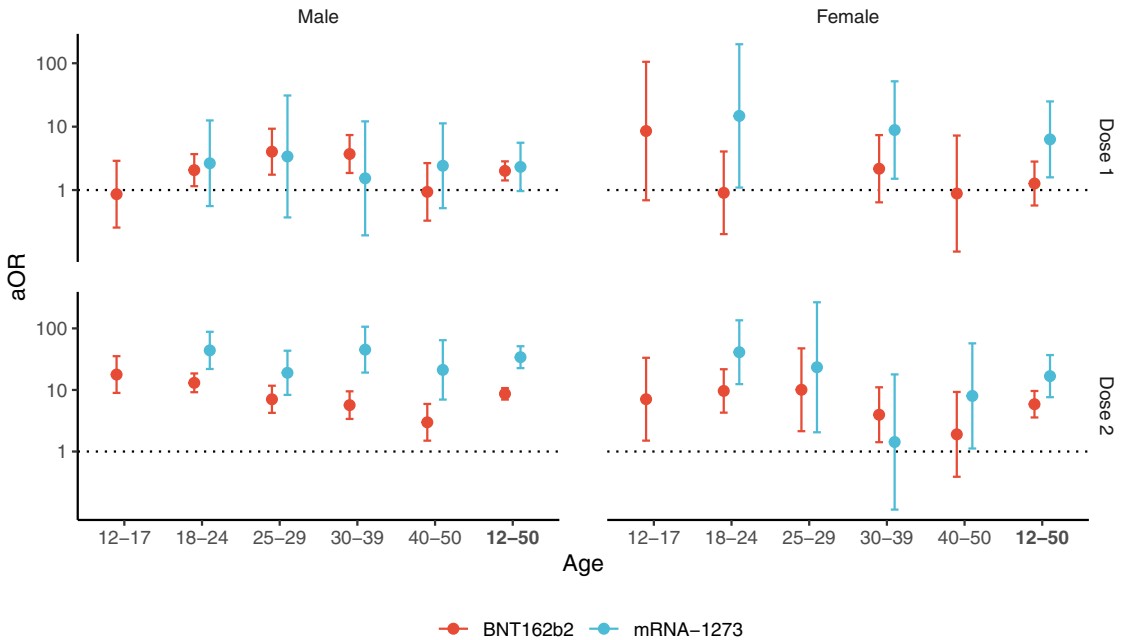

**Fig. 1 Association between myocarditis and exposure to mRNA vaccines within 7 days, according to sex and age group.** Adjusted odds-ratio (aOR) from multivariable model are represented in base 10 logarithmic scale according to age groups (x-axis), by sex (columns) and vaccine dose ranking (rows). Colors denote the type of vaccine. Centre value are aOR point estimates and error bars represent 95% confidence intervals. Number of cases (N) by age categories (12-17, 18-24, 25-29, 30-39, 40-50 and 12-50 years) are respectively as follows: N = 137, 480, 210, 273, 181 and 1281 for males, and N = 29, 106, 40, 88, 68 and 331 for females. aOR could not be calculated in categories where no case exposed to vaccine was recorded, for instance for males and females aged 12 to 17 years having received the mRNA-1273 vaccine.

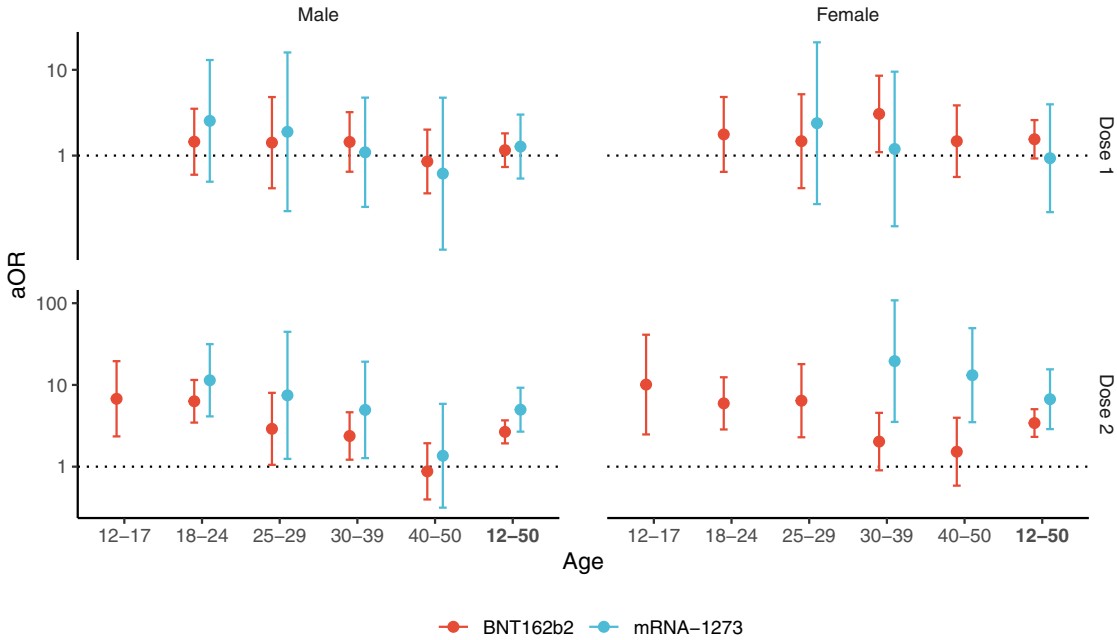

**Fig. 2 Association between pericarditis and exposure to mRNA vaccines within 7 days, according to sex and age group.** Adjusted odds-ratio (aOR) from multivariable model are represented in base 10 logarithmic scale according to age groups (x-axis), by sex (columns) and vaccine dose ranking (rows). Colors denote the type of vaccine. Centre value are aOR point estimates and error bars represent 95% confidence intervals. Number of cases (N) by age categories (12-17, 18-24, 25-29, 30-39, 40-50 and 12-50 years) are respectively as follows: N = 65, 194, 106, 282, 342 and 989 for males, and N = 36, 118, 91, 183, 196 and 624 for females. aOR could not be calculated in categories where no case exposed to vaccine was recorded, for instance for males and females aged 12 to 17 years having received the mRNA-1273 vaccine.

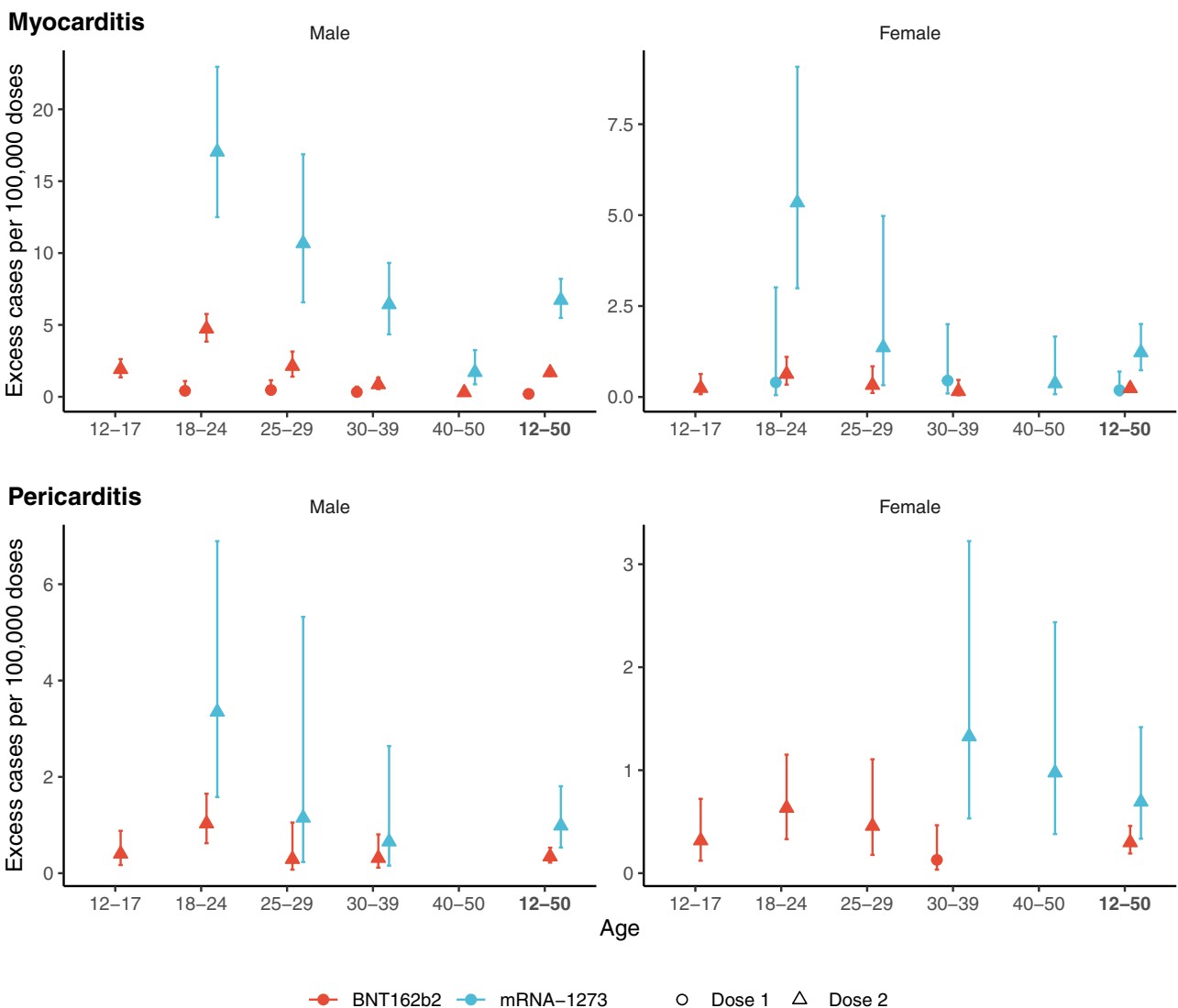

**Fig. 3 Excess cases of myocarditis and pericarditis attributable to mRNA vaccines according to sex and age group, per 100,000 doses.** Excess cases are based on the risk in the 7 days following vaccination. Colors denote the type of vaccine and the shape of point estimate denotes the ranking of dose vaccine. Centre value are excess cases point estimates and error bars represent 95% confidence intervals. Number of cases (N) by age categories (12-17, 18-24, 25-29, 30-39, 40-50 and 12-50 years) are respectively as follows: for cases of myocarditis, N = 137, 480, 210, 273, 181 and 1281 in males, and N = 29, 106, 40, 88, 68 and 331 in females; for cases of pericarditis, N = 65, 194, 106, 282, 342 and 989 in males, and N = 36, 118, 91, 183, 196 and 624 in females. Excess cases was only calculated in categories with a significantly positive association between the vaccine exposure and the outcome (adjusted odds-ratio >1).

myocarditis (none among exposed to vaccine) and 5 (0.31%) deaths among cases of pericarditis (including one patient having received a vaccine 8 to 21 days prior to the diagnosis) were reported. Of those, 3 and 2 died during their hospital stay for myocarditis and pericarditis, respectively.

Drugs treatments within 30 days after hospital discharge are presented in Figs. S3 and S4. Regardless of the vaccination status, the therapeutic classes most frequently used during the follow-up of myocarditis cases included beta blocking agents (63% of patients), analgesics (52%) and agents acting on the renin −angiotensin system (46%). The corresponding treatments of pericarditis cases were analgesics (83%), colchicine (69%) and beta blocking agents (14%) (Fig. S4).

## Discussion

In this nationwide study involving a population of 32 million people aged 12 to 50 years having received 46 million doses of

mRNA vaccines, we provide detailed estimates of the risk of myocarditis and pericarditis by sex, age categories and vaccine type. We find that vaccination with both mRNA vaccines was associated with an increased risk of myocarditis and pericarditis within the first week after vaccination. The associations were particularly pronounced after the second dose, and were evident in both males and females. We found a trend of increased risks towards younger age groups but a significant risk was also found in males over 30 years to develop myocarditis and in females over 30 years to develop a pericarditis after vaccination. Reassuringly, these cases of myocarditis and pericarditis, although requiring hospitalization, did not result in more severe outcomes than those unrelated to vaccination.

Our results are generally consistent with those reported by the pharmacovigilance systems in France and other countries[8,13–16]. Several common factors in terms of the characteristics and prognosis of cases identified, and the temporal relationship

**Table 3 Description of hospitalized patients according to the exposure to mRNA vaccines.**

| | Myocarditis | | | Pericarditis | | |
|---|---|---|---|---|---|---|
| | Unexposed | Vaccinated within 1 to 7 days | Vaccinated within 8 to 21 days | Unexposed | Vaccinated within 1 to 7 days | Vaccinated within 8 to 21 days |
| | (N = 1077) | (N = 378) | (N = 157) | (N = 1267) | (N = 172) | (N = 174) |
| Sex | | | | | | |
| Male | 829 (77.0) | 324 (85.7) | 128 (81.5) | 778 (61.4) | 101 (58.7) | 110 (63.2) |
| Female | 248 (23.0) | 54 (14.3) | 29 (18.5) | 489 (38.6) | 71 (41.3) | 64 (36.8) |
| Age[a] | | | | | | |
| Mean (sd) | 28.5 (9.74) | 25.6 (8.44) | 28.6 (9.53) | 33.8 (10.3) | 29.9 (10.0) | 33.9 (10.0) |
| Median (range) | 26.0 (21.0–36.0) | 23.0 (19.0–30.8) | 26.0 (20.0–37.0) | 35.0 (25.0–43.0) | 29.0 (21.0–38.0) | 34.0 (26.0–42.0) |
| Age distribution[a] | | | | | | |
| 12–17 | 114 (10.6) | 40 (10.6) | 12 (7.6) | 80 (6.3) | 12 (7.0) | 9 (5.2) |
| 18–24 | 356 (33.1) | 171 (45.2) | 59 (37.6) | 228 (18.0) | 56 (32.6) | 28 (16.1) |
| 25–29 | 168 (15.6) | 60 (15.9) | 22 (14.0) | 152 (12.0) | 22 (12.8) | 23 (13.2) |
| 30–39 | 248 (23.0) | 74 (19.6) | 39 (24.8) | 361 (28.5) | 47 (27.3) | 57 (32.8) |
| 40–50 | 191 (17.7) | 33 (8.7) | 25 (15.9) | 446 (35.2) | 35 (20.3) | 57 (32.8) |
| Deprivation Index[b] | | | | | | |
| Most deprived | 654 (60.7) | 237 (62.7) | 95 (60.5) | 820 (64.7) | 115 (66.9) | 114 (65.5) |
| Least deprived | 423 (39.3) | 141 (37.3) | 62 (39.5) | 447 (35.3) | 57 (33.1) | 60 (34.5) |
| History of myocarditis or pericarditis[c] | 104 (9.7) | 12 (3.2) | 10 (6.4) | 149 (11.8) | 10 (5.8) | 14 (8.0) |
| History of SARS-CoV-2 infection[d] | 58 (5.4) | 2 (0.5) | 4 (2.5) | 39 (3.1) | 0 (0) | 3 (1.7) |
| Length of hospital stay | | | | | | |
| Mean (sd) | 4.56 (5.97) | 3.75 (2.60) | 4.18 (2.70) | 2.84 (4.46) | 2.36 (2.49) | 2.52 (2.84) |
| Median (range) | 4.00 (2.00–5.00) | 4.00 (2.00–5.00) | 4.00 (3.00–5.00) | 1.00 (0–4.00) | 2.00 (1.00–4.00) | 2.00 (1.00–3.00) |
| Death up to 30 days after discharge | 4 (0.4) | 0 (0) | 0 (0) | 4 (0.3) | 0 (0) | 1 (0.6) |
| among which deceased during hospital stay | 3 (0.3) | 0 (0) | 0 (0) | 1 (0.1) | 0 (0) | 1 (0.6) |
| Intensive care unit | 66 (6.1) | 9 (2.4) | 6 (3.8) | 32 (2.5) | 0 (0) | 2 (1.1) |
| Ventilation - oxygen therapy | 46 (4.3) | 12 (3.2) | 5 (3.2) | 30 (2.4) | 1 (0.6) | 3 (1.7) |
| Pericardial drainage | 3 (0.3) | 0 (0) | 0 (0) | 38 (3.0) | 1 (0.6) | 2 (1.1) |

[a]At index date (date of hospital admission for myocarditis or pericarditis).
[b]Least deprived refers to the grouping of 1st and 2nd quintiles, and most deprived to the grouping of 3d to 5th quintiles of the deprivation index.
[c]Defined as an hospitalization with the respective condition within past 5 years.
[d]Either a positive RT-PCR or antigenic test for SARS-CoV-2, or hospitalization for COVID-19, within 30 days prior to index date.

between vaccine exposure and the event of interest, suggest a consistent underlying mechanism[5,6,17,18]. As found in our analyses, various reports indicate that the risk is more pronounced with the mRNA-1273 vaccine[7,10,19,20], even though there was no difference in rates between the two vaccines in the passive surveillance reporting in the US[4].

Our findings bring new elements in showing that the risk of acute cardiac inflammation after vaccination is not confined to myocarditis in young men[4–6,14]. First, in line with results from a cohort study in Nordic countries[11], our analyses show a significant risk and population burden of pericarditis following the second dose of the BNT162b2 and mRNA-1273 vaccine. Often comprised in a combined outcome of myopericarditis[7,19,21], pericarditis as specific entity has been less studied for its association with mRNA vaccines, and even more rarely regarding the mRNA-12173 vaccine. For the BNT162b2 vaccine, results are inconsistent with either reports of a positive association[11,18] or an absence of association[8–10]. Barda et al. and Lai et al. found a non-significant risk ratio of 1.27 and odds ratio of 1.06, respectively, for the combined effect of first and second dose of the BNT162b2 vaccine[8,9]. Patone et al. found a non-significant relative incidence of pericarditis in the week after both doses of the BNT162b2 vaccine of approximately 0.6, while the association with mRNA-1273 could not be quantified[10]. Considering that the risk of

myocarditis following the BNT162b2 vaccine is also found lower in the later study than in others, we hypothesized that the probably weaker association with pericarditis might be more difficult to reveal. This discrepancy could also reflect different diagnostic practices as pericarditis is a retrospective diagnosis of exclusion.

Second, by differentiating the risk between adolescent (aged 12 to 17 years) and young men or women (18–25 years), we estimate that the number of excess cases after the second dose of BNT162b2 vaccine is lower in adolescents compared to young adults. This is consistent with findings from surveillance data in Israel[22] but in contrast with those from the US[4]. There is some support for the role of sex hormones in the increased susceptibility for myocarditis of young men compared to women[23–25]. While we do find higher absolute burden of myocarditis and pericarditis in adolescent males and men, we also find that the female counterpart also faces a significant risk, notably of pericarditis for women over 30 years after the second dose of the mRNA-1273 vaccine, which has not yet been shown.

There are several factors that support the hypothesis of a causal relationship between exposure to mRNA vaccines and the risk of myocarditis and pericarditis. First, the associations remained strong, even after adjusting for a history of these conditions or

recent SARS-CoV-2 infection, and in a period during which most common respiratory viruses were not widely circulating[26,27]. Second, the time that elapsed between exposure to the vaccine and hospitalization was very short for both conditions, particularly after the second dose. Third, in most cases, the associations did not persist after seven days following exposure. Fourth, the stronger risk associated with the second dose and the mRNA-1273 vaccine, which contains a larger amount of mRNA, suggest a dose response relationship[28].

The strengths of our study include the large sample size, population-based character and the assessment of cases and exposure to vaccines in high-quality and comprehensive databases. It allowed us to include 1612 confirmed cases of myocarditis and 1613 of pericarditis, occurring in a period during which 46 million doses of the two mRNA vaccines were administered. This study provides population estimates of vaccine associated risk and burden at a national level, which cannot be informed by passive case notification surveillance. Furthermore, results were consistent after adjusting for other risk factors, including SARS-CoV-2 infection, and different periods.

Our study has several limitations. First, the National Health Data System provides little clinical and no laboratory information concerning cases. The cases included in this study were identified solely on the basis of the diagnosis codes associated with hospital admissions. We therefore could not detect asymptomatic or mild forms of myocarditis and pericarditis that would not require hospitalization. Nevertheless, the incidence of myocarditis and pericarditis before the Covid-19 pandemic, estimated using the SNDS data, is consistent with the figures reported by other countries[14]. Furthermore, the observed durations of stay and post-discharge treatments were consistent with typical presentations of these conditions. Second, while our assessment of severity indicators within four weeks post-discharge indicates a favourable clinical outcome of post-vaccination carditis in their acute phase, we could not investigate potential long-term consequences. Third, we did not study the Covid-19 booster vaccination which was not yet recommended for healthy younger adults in our study period. Finally, associations across age and sex subgroups could not always be quantified for both vaccines or only with a considerable degree of uncertainty due to the limited time span of observation. The extent of the risk for certain subgroups, especially among women, for whom the incidence appears to be lower, warrants further studies and meta-analyses[26,29].

In conclusion, this study provides strong evidence of an increased risk of myocarditis and of pericarditis in the week following vaccination against Covid-19 with mRNA vaccines in both males and females, in particular after the second dose of the mRNA-1273 vaccine. Future studies based on an extended period of observation will allow to investigate the risk related to the booster dose of the vaccines and monitoring the long-term consequences of these post vaccination acute inflammations.

## Methods

**Study design**. We conducted a matched case-control study within the entire French population between 12 and 50 years of age for myocarditis and pericarditis, treating each condition separately. The study focused on the period from May 12, 2021, to October 31, 2021, during which the Covid-19 vaccination campaign was opened to individuals under 50 years of age.

**Data sources and study population**. The study was based on data of the National Health Data System (SNDS) which covers more than 99% of the French population (67 million inhabitants)[30,31]. Data on hospital admission were obtained from the French hospital discharge database (PMSI) and linked at the individual level with the nationwide databases for Covid-19 vaccination (VAC-SI) and testing (SI-DEP). Cases corresponded to all patients admitted to French hospitals with a diagnosis of myocarditis or pericarditis in the study period. Diagnoses at hospital were typically based on presenting symptoms, electrocardiography, echocardiography and cardiac

magnetic resonance imaging[32,33]. We used the codes for myocarditis (I40.x, I41.x, and I51.4) and pericarditis (I30.x and I32.x) of the International Classification of diseases, 10th revision (ICD-10) for detection. Although the data were comprehensive up to September 2021, at the time this study was conducted, approximately 78% of hospital stays for October 2021 had been entered into the PMSI database. Each case was matched at the date of his/her hospital admission for myocarditis or pericarditis (index date) to 10 control individuals. Controls were selected from among the whole population by simple random sampling without replacement within each stratum of age, gender and area of residence (matching criteria), with constraint of not being diagnosed with myocarditis or pericarditis and being alive at the index date.

Our research group (EPI-PHARE) has a regulatory permanent access to the data from the SNDS. This permanent access is given according the French Decree No. 2016-1871 of December 26, 2016 relating to the processing of personal data called "National Health Data System" and French law articles Art. R. 1461-13 and 14. This study was declared prior to initiation on the EPI-PHARE registry of studies requiring the use of the SNDS (n° EP-0311). No informed consent was required because data are anonymized.

**Exposure and covariates**. Exposure was defined as vaccination with an mRNA vaccine 1 to 7 days or 8 to 21 days prior to the index date, considering the first and second dose separately. Non-vaccinated subjects, and those vaccinated more than 21 days before the index date were considered to be non-exposed. In addition to the matching variables, three covariates potentially associated with a risk of myocarditis or pericarditis, and with vaccine exposure were considered. A prior history of myocarditis or pericarditis was defined as a hospital admission with an ICD-10 code for myocarditis or pericarditis (cf. above) in the five years preceding the index date. A history of severe acute respiratory syndrome coronavirus 2 (SARS-CoV-2) infection was defined by hospital admission for Covid-19 or a positive polymerase-chain-reaction (PCR) or antigenic testing 30 days prior to the index date. The socio-economic level was defined by a deprivation index, summarized in two categories[34].

**Statistical analysis**. We used conditional logistic regression models to estimate the odds ratios (OR) of myocarditis and pericarditis associated with exposure to recent vaccination, adjusted for covariates and matching variables[35]. Analyses were conducted with reference to the ranking of vaccine dose (first or second dose) and the time elapsed since vaccination (1 to 7 days or 8 to 21 days), across the study group as a whole and separately for males and females and according to five age brackets (12–17, 18–24, 25–29, 30–39 and 40 to 50 years). Associations were measured relative to the most recent exposure. We estimated the number of cases attributable to vaccine exposure using the odds ratio as an estimate of relative risk and assuming a causal relationship[36]. We then derived two measures of population burden using information on exposure to vaccines across the 32.2 million people aged 12 to 50 years, including the vaccine type and date of receipt of each dose (Table S1). First, the number of doses required for the occurrence of a vaccine-associated case was estimated as the ratio of doses administered to the number of attributable cases. Second, the number of excess cases per 100,000 doses was derived by inverting this ratio. We applied a correction factor to the numbers of exposed cases to account for under-reporting of hospitalizations in October 2021. Confidence intervals for the number of cases attributable to exposure were obtained by application of the delta-method[37,38]. We assessed the sensitivity of the results to a potential ascertainment bias by performing an analysis restricted to the time period before July 19, 2021, i.e. before myocarditis and pericarditis were officially announced as adverse events of mRNA vaccines. Additional analyses were conducted by excluding (i) patients with a previous history of myocarditis or pericarditis, (ii) those with a history of SARS-CoV-2 infection, (iii) patients having both diagnoses of myocarditis and pericarditis, and (iv) persons with a hospitalization within 28 days of index date. Data collection used SAS Enterprise Guide version 4.3 software (SAS Institute, Cary, North Carolina). All analyses were performed using R software version 4.1.3, and survival package version 3.2–13[39,40].

**Reporting summary**. Further information on research design is available in the Nature Research Reporting Summary linked to this article.

## Data availability

According to data protection and the French regulation, the authors cannot publicly release the data from the French national health data system (SNDS). However, any person or structure, public or private, for-profit or non-profit, is able to access SNDS data upon authorization from the French Data Protection Office (CNIL Commission Nationale de l'Informatique et des Libertés) to carry out a study, a research, or an evaluation of public interest (https://www.snds.gouv.fr/SNDS/Processus-d-acces-aux-donnees and https://www.indsante.fr/).

## Code availability

The code to reproduce the analyses presented in the paper is publicly available[41].

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

## Author contributions

S.L.V., M.B., A.W., R.D.S. and M.Z. conceived the study. M.B. and J.D. supervised the project. A.W., R.D.S. and M.Z. carried out the clinical data collection and data curation. S.L.V. and M.B. designed and performed the statistical analyses with M.J.J., B.B. and J.B. providing input. S.L.V. wrote the first draft of the manuscript. All authors interpreted the results, provided critical revision of the manuscript and approved its final version for submission.

## Competing interests

The authors declare no competing interests.
