## [Peer Review File · Nature Communications]

Age and sex-specific risks of myocarditis and pericarditis following COVID-19 messenger RNA VaccinesREVIEWER COMMENTS

Reviewer #1 (Remarks to the Author):

The article by Stéphane Le Vu and colleagues is well-written, well-analyzed, and very relevant. The large size of the material makes it an important piece of evidence describing the association between SARS-CoV-2 mRNA vaccines and myocarditis/pericarditis.

I think the paper deserves publication after one round of minor revisions:

Major comments:

p14 line 285-288: The only drawback of this study is that the study does not describe in detail how cases and controls were matched 'randomly' by age, gender, and area. Which process/technique was used to do this? And if whole-population information on the French population was available, why was the study not conducted as cohort study?

Minor comments:

p10 line 202-204: Alternatively, this could reflect different diagnostic practices as pericarditis typically is a diagnosis in the absence of other diagnoses.

p10 line 218-219: Not sure this statement is completely true, as far as I know rhinovirus was prevalent during this period.

p11 line 222-224: This sentence mixes the effects of vaccine dosage (i.e., microgram of mRNA) and the vaccine dose number (i.e., first dose, second dose, ect.). Given that the myocarditis risk does not seem to be markedly higher for boosters (3rd dose), I would consider rewriting this sentence as it is not a simple dose-response relationship for both vaccine dosage and vaccine dose.

Thank you for writing such a good and highly relevant manuscript.

Reviewer #2 (Remarks to the Author):

Thank you for the opportunity to review this interesting nested case-control study on the association of the first two doses of two mRNA COVID-19 vaccines with myocarditis and pericarditis in the French population. Both vaccines were found significantly associated with the two conditions, within the first week after both doses of both vaccines for myocarditis and the first week after the second dose of both vaccines for pericarditis. The sample size is much larger than a previous study of a similar design in *Annals of Internal Medicine* looking into an inactivated vaccine and an mRNA vaccine (<http://doi.org/10.7326/M21-3700>). Methods seem sound. Findings are largely consistent with the established knowledge about the association and provide a more comprehensive assessment of this association by first/second dose which I think may merit publication subject to satisfactory revisions addressing the following specific issues.

1. By reading the description of the design I reckon this is a nested case-control study and is derived from an underlying cohort. However, the cohort is not sufficiently described, such that the pool from which the control patients were selected is not well characterized. Who in France have their records kept in this database? Everyone? Even those without a previous disease or admission? If not, it should be made clear the population of this study are healthcare service users only and might be slightly limited in the generalizability to the general population.

2. Another question regarding the data source, do you have outmigration records? If you do probably you should use outmigration as one of the observation endpoints (for controls) for the matching as well although it would likely have minor effect on the results.

3. Were you able to adjust for ethnicity? If so why did you not?

4. Myocarditis is sometimes misdiagnosed as myocardial infarction or the other way around is possible too. Would you consider any sensitivity analysis in the view of this possibility?

Misdiagnosis and reporting bias can be a major problem as clinicians and patients are well aware of the potential myopericarditis in mRNA vaccine recipients. Hence we cannot exclude the possibility of misdiagnosis (mislabel) of certain symptoms such as chest pain as myopericarditis in the mRNA vaccine recipients. Since cardiac enzymes, cardiac echogram, ECG and cMRI etc are not available, the authors will not be able to evaluate this potential confounder.

5. This study did not restrict cases to only first diagnosis of myocarditis/pericarditis. By adopting this approach, I think the cases might not be exactly what you would like to investigate because your hypothesis is vaccination alone can induce myocarditis. You have conducted a sensitivity analysis for first diagnosis only (five-year wash out) showing no substantial differences in the findings which is good but have you tried looking at whether there is an interaction between vaccination and myocarditis/pericarditis history? I think it'd be useful information for those who had such a history when considering vaccination: whether they have the same risk as everybody else? Or even more?

6. The method says: "Each case was matched at the date of his/her hospital admission for myocarditis or pericarditis (index date) to 10 control individuals, randomly selected from among the whole population, not diagnosed with myocarditis or pericarditis and still alive at the index date. Matching criteria were age, gender and the area of residence (i.e. 101 départements, including overseas territories)." If the controls were already in the hospital before the index date then the chance of having vaccination may be changed and the likelihood of having vaccination 28 days before the index date will be lower. It may artificially inflated the odds ratio. I suggest removing all the controls already in the hospitals within 28 days prior to the index date.

7. For such a large sample size I think more covariates can be included. Can I ask why not more covariates such as heart failure history were included? Why are drug use within the past 90 days not considered for inclusion?

8. The cases are only defined using ICD-10 diagnostic codes. Were there any protocols for the diagnosis of myocarditis/pericarditis in the hospitals for clinicians? Brighton Collaboration Criteria? Maybe useful to describe the procedures or criteria used to diagnose the conditions since no laboratory data is available for you to further remove unlikely cases to make sure the validity of the cases. Or is the whole database validated for accuracy?

9. I am not sure if adjustment is the best way to deal with SARS-CoV-2 infection. I would have used it as one of the observation endpoints for the underlying cohort and exclude those who already had an infection prior to the index date. This would make it more purely about the safety of the vaccine itself.

Reviewer #3 (Remarks to the Author):

The manuscript entitled "Age and sex-specific risks of myocarditis and pericarditis following Covid-19 messenger RNA Vaccines: a nationwide assessment" by Vu et al., aimed to estimate risks of myocarditis and pericarditis by Covid-19 vaccine type (Pfizer and Moderna) and across sex and age groups, using a matched case-control study design (a separate one for each outcome). Information was collected through linkage of the French National Health Data System with the Covid-19 vaccination database. The results indicated increased risks of both myocarditis (mostly in younger people, and in males and females) and pericarditis (mostly in older subjects) during the first week following both the first and the second vaccine doses, for both vaccine types. Age and sex-specific estimates of excess cases attributable to vaccination revealed a substantial burden of both myocarditis and pericarditis.

GENERAL COMMENTS

The use of administrative data in a medical research present some inherent disadvantages, some of which are addressed by the authors in the 'Discussion' section. However, the results of the study confirm previous findings and add some new observations, for example regarding perimyocarditis risk in older subjects (less studied so far) as well as myocarditis risks in young females, not only males. COVID-19 mRNA vaccines are still "young", and therefore gathering information about their potential adverse effects is of high importance.

SPECIFIC COMMENTS

Methods -

Please comment on the completeness of both databases that were used.

Were outcomes included only if occurred up to 21 days following vaccination? that makes sense for the Pfizer vaccine, where the gap between the two doses was usually of 21 days, but for the Moderna vaccine, the gap was at least 28 days, so in theory a later outcome could have also been included. Where there cases of people vaccinated with both vaccine types? how were they treated?

or was that an exclusion criterion?

Discussion -

Please comment on the accuracy of the ICD-10 codes for myocarditis and pericarditis in the database used (reproducibility, validity), since this could be a source for outcome misclassification, which may be differential.

Response to referees for the article “Age and sex-specific risks of myocarditis and pericarditis following Covid-19 messenger RNA Vaccines: a nationwide assessment.” submitted to Nature Communications.

Dear Editor,

Thank you for the opportunity to revise our paper. Our responses are interspersed with reviewers’ comments in **blue arial font**. We use *Italic type* to indicate changes in the edited version. The changes are marked in “track changes” mode in the revised manuscript.

Sincerely,

Stéphane Le Vu, on behalf of all co-authors

Reviewer #1 (Remarks to the Author):

The article by Stéphane Le Vu and colleagues is well-written, well-analyzed, and very relevant. The large size of the material makes it an important piece of evidence describing the association between SARS-CoV-2 mRNA vaccines and myocarditis/pericarditis. I think the paper deserves publication after one round of minor revisions:

Major comments:

p14 line 285-288: The only drawback of this study is that the study does not describe in detail how cases and controls were matched 'randomly' by age, gender, and area. Which process/technique was used to do this? And if whole-population information on the French population was available, why was the study not conducted as cohort study?

Controls were selected by stratified simple random sampling without replacement. We edited the corresponding section to clarify this:

“Each case was matched at the date of his/her hospital admission for myocarditis or pericarditis (index date) to 10 control individuals. Controls were selected from among the whole population by simple random sampling without replacement within each stratum of age, gender and area of residence (matching criteria), with constraint of not being diagnosed with myocarditis or pericarditis and being alive at the index date.”

We chose a case-control study design over a cohort, to avoid the computational burden of individually matching a very large population unexposed and exposed subjects, for outcomes that were deemed rare, and to allow for different exposure windows under consideration.

Minor comments:

p10 line 202-204: Alternatively, this could reflect different diagnostic practices as pericarditis typically is a diagnosis in the absence of other diagnoses.

Thank you for pointing out this explanation. We added it in the discussion:

“Considering that the risk of myocarditis following the BNT162b2 vaccine is also found lower in the later study than in others, we hypothesized that the probably weaker association with pericarditis might be more difficult to reveal. This discrepancy could also reflect different diagnostic practices as pericarditis is a retrospective diagnosis of exclusion.”

p10 line 218-219: Not sure this statement is completely true, as far as I know rhinovirus was prevalent during this period.

Indeed, the diffusion of rhinovirus seems to be the least affected by mitigation measures against Covid-19. A decline in positivity rates – compared to prepandemic era – has been observed for the other common respiratory viruses, influenza, parainfluenza, metapneumovirus, human coronaviruses, adenovirus and respiratory syncytial virus.

In France, rates of acute respiratory infections and bronchiolitis stayed at low levels prior to November 2021 (Launay et al. <https://doi.org/10.1371/journal.pone.0258391> - Réseau Sentinelles <https://www.sentiweb.fr/france/fr/?page=table&maladie=25>). Thus, seasonal infections – including rhinovirus which appears rarely associated with myocarditis (Chow et al.. <https://doi.org/10.1016/j.idcr.2020.e00702>) – are unlikely to explain the absolute excess of myo-pericarditis we measure during the vaccination period from May to October 2021.

Nevertheless, to tone down our statement, we changed this sentence to:

“... in a period during which most common respiratory viruses were not widely circulating”.

p11 line 222-224: This sentence mixes the effects of vaccine dosage (i.e., microgram of mRNA) and the vaccine dose number (i.e., first dose, second dose, ect.). Given that the myocarditis risk does not seem to be markedly higher for boosters (3rd dose), I would consider rewriting this sentence as it is not a simple dose-response relationship for both vaccine dosage and vaccine dose.

We concede that our results would not be able to support more than a suggestion, but there is also evidence of a cumulative effect of doses, including the booster (see the preprint from Patone et al. (2021). <https://doi.org/10.1101/2021.12.23.21268276>). Also, the delay between exposure to the second and booster doses (half dose for the mRNA-1273 vaccine) might also come into play to explain the relatively smaller effect.

Thank you for writing such a good and highly relevant manuscript.

Thank you for your assessment and positive comments.

Reviewer #2 (Remarks to the Author):

Thank you for the opportunity to review this interesting nested case-control study on the association of the first two doses of two mRNA COVID-19 vaccines with myocarditis and pericarditis in the French population. Both vaccines were found significantly associated with the two conditions, within the first week after both doses of both vaccines for myocarditis and the first week after the second dose of both vaccines for pericarditis. The sample size is much larger than a previous study of a similar design in *Annals of Internal Medicine* looking into an inactivated vaccine and an mRNA vaccine (<http://doi.org/10.7326/M21-3700>). Methods seem sound. Findings are largely consistent with the established knowledge about the association and provide a more comprehensive assessment of this association by first/second dose which I think may merit publication subject to satisfactory revisions addressing the following specific issues.

Thank you for your thorough assessment and insightful comments.

1. By reading the description of the design I reckon this is a nested case-control study and is derived from an underlying cohort. However, the cohort is not sufficiently described, such that the pool from which the control patients were selected is not well characterized. Who in France have their records kept in this database? Everyone? Even those without a previous disease or admission? If not, it should be made clear the population of this study are healthcare service users only and might be slightly limited in the generalizability to the general population.

All adults residing in France have a national health insurance card and a unique identifier, either attributed at birth or at the time of legal immigration. The SNDS database contains records of all medical expenses individually reimbursed and hospital stays in private or public hospitals since 2006. This leaves very few persons out of the databases, and more than 99% of the population is reported to be covered [Bezin et al. <https://doi.org/10.1002/pds.4233> and Semenzato et al. <https://doi.org/10.1016/j.lanepe.2021.100158>].

We added these two references and mentioned the coverage in the methods section.

2. Another question regarding the data source, do you have outmigration records? If you do probably you should use outmigration as one of the observation endpoints (for controls) for the matching as well although it would likely have minor effect on the results.

We don't have this information. As noted, we think that the situation where a control could experience the outcome, within 3 weeks of vaccination, which would stay undetected because of migration has very little chance occur, and is unlikely to affect the results.

3. Were you able to adjust for ethnicity? If so why did you not?

We were not able to use this type of information as there is no record of ethnicity in administrative data in France. Besides, even though US studies typically report

myocarditis cases with a breakdown by race/ethnicity, authors do not seem to note a specificity in occurrence rates.

4. Myocarditis is sometimes misdiagnosed as myocardial infarction or the other way around is possible too. Would you consider any sensitivity analysis in the view of this possibility? Misdiagnosis and reporting bias can be a major problem as clinicians and patients are well aware of the potential myopericarditis in mRNA vaccine recipients. Hence we cannot exclude the possibility of misdiagnosis (mislabel) of certain symptoms such as chest pain as myopericarditis in the mRNA vaccine recipients. Since cardiac enzymes, cardiac echogram, ECG and cMRI etc are not available, the authors will not be able to evaluate this potential confounder.

We tried to address this in two ways. First, for detecting a reporting bias, we performed an analysis restricted to the period prior to the official warning against myocarditis as a side effect of mRNA vaccine, and found similar associations. Second, to assess the specificity of the diagnoses, we gathered post discharge drug treatments that appear typical of myocarditis and pericarditis, with for instance no statins that would sign a treatment of myocardial infarction.

We would also like to emphasize two other points that lessen the potential confounding with myocardial infarction or other cardiovascular conditions:

(1) our study population is relatively young (median of 25 years for the myocarditis analysis and 34 for pericarditis),

(2) diagnosis codes are collected and validated to determine the cost of each hospital stay, and several validation studies have shown a high specificity of the classification of cardiovascular diseases (<https://doi.org/10.1007/s00415-012-6686-0>, <https://doi.org/10.1002/pds.4690>, <https://doi.org/10.1159/000511206>, <https://doi.org/10.1159/000438859>, <https://doi.org/10.1016/j.acvd.2016.01.012>, <https://doi.org/10.1111/fcp.12326>).

5. This study did not restrict cases to only first diagnosis of myocarditis/pericarditis. By adopting this approach, I think the cases might not be exactly what you would like to investigate because your hypothesis is vaccination alone can induce myocarditis. You have conducted a sensitivity analysis for first diagnosis only (five-year wash out) showing no substantial differences in the findings which is good but have you tried looking at whether there is an interaction between vaccination and myocarditis/pericarditis history? I think it'd be useful information for those who had such a history when considering vaccination: whether they have the same risk as everybody else? Or even more?

Thank you for suggesting this.

We tested the interaction between mRNA vaccine exposure (V) and myocarditis/pericarditis history (H), along with other covariates (X, SARS-CoV-2 infection and deprivation index), as predictors of myocarditis (Y) in a model of the form:

$$\text{logit}(P(Y = 1)) = V + H + V:H + X + \text{strata}$$

And found no significant interaction (V:H term):

	coef	exp(coef)	se(coef)	z	p
V	1.06328	2.89584	0.06353	16.737	<2e-16
H	5.27171	194.74918	0.39270	13.424	<2e-16
X1	2.09692	8.14105	0.17030	12.313	<2e-16
X2	-0.12375	0.88360	0.06621	-1.869	0.0616
V:H	-1.20250	0.30044	0.84042	-1.431	0.1525

We thus added this sentence in the results section:

“No interaction was found between history of carditis and vaccine exposure.”

6. The method says: “Each case was matched at the date of his/her hospital admission for myocarditis or pericarditis (index date) to 10 control individuals, randomly selected from among the whole population, not diagnosed with myocarditis or pericarditis and still alive at the index date. Matching criteria were age, gender and the area of residence (i.e. 101 départements, including overseas territories).” If the controls were already in the hospital before the index date then the chance of having vaccination may be changed and the likelihood of having vaccination 28 days before the index date will be lower. It may artificially inflated the odds ratio. I suggest removing all the controls already in the hospitals within 28 days prior to the index date.

Thank you for raising this point, we have taken your advice. Controls reported to have been hospitalized within 28 days of index date represented 0.4% for myocarditis and 0.6% for the analysis of pericarditis. As suggested, we ran a model excluding these individuals. The adjusted odds ratio are virtually unchanged relative to the main results, and are now represented in figure S1 (model F) and mentioned in the results section.

7. For such a large sample size I think more covariates can be included. Can I ask why not more covariates such as heart failure history were included? Why are drug use within the past 90 days not considered for inclusion?

Aside from the matching variables (age, gender, area of residence), we included three covariates used for adjustment (previous history of myocarditis or pericarditis, history of SARS-CoV-2 infection, and social deprivation). We chose these adjustment variables on the basis of the initial reports of vaccine-associated carditis that described young patients without comorbidities. (Bozkurt et al. <https://doi.org/10.1161/CIRCULATIONAHA.121.056135>). Our population sample is indeed relatively young (median of 25 years in the myocarditis analysis), such that the prevalence of chronic condition or drug use should be limited.

8. The cases are only defined using ICD-10 diagnostic codes. Were there any protocols for the diagnosis of myocarditis/pericarditis in the hospitals for clinicians? Brighton Collaboration Criteria? Maybe useful to describe the procedures or criteria used to diagnose the conditions

since no laboratory data is available for you to further remove unlikely cases to make sure the validity of the cases. Or is the whole database validated for accuracy?

We added the following sentence in the methods section to describe the current diagnostic practice in France:

“Diagnoses at hospital were typically based on presenting symptoms, electrocardiography, echocardiography and cardiac magnetic resonance imaging [Adler <https://doi.org/10.1093/eurheartj/ehv318>, Hékimian <https://doi.org/10.1007/s13546-017-1273-4>].”

There are several reports of (good) accuracy of ICD-10 coding in our database for other cardiovascular diseases : heart failure (Bosco-Lévy et al. <https://doi.org/10.1002/pds.4690>), thrombotic events (Prat et al. <https://doi.org/10.1111/fcp.12326>) and ischaemic stroke (Aboa-Eboulé et al. <https://doi.org/10.1007/s00415-012-6686-0> and Giroud et al. <https://doi.org/10.1159/000438859>), although not specifically for myo/pericarditis.

Again, doctors specialised in medical information collect and validate the coding from clinicians, for financing and controlling purposes.

9. I am not sure if adjustment is the best way to deal with SARS-CoV-2 infection. I would have used it as one of the observation endpoints for the underlying cohort and exclude those who already had an infection prior to the index date. This would make it more purely about the safety of the vaccine itself.

We did use SARS-CoV-2 infection as an adjustment, but also performed an analysis completely excluding patients with history of SARS-CoV-2 infection (as represented in in figure S1, model C) and found no substantial difference in the OR estimates.

Reviewer #3 (Remarks to the Author):

The manuscript entitled "Age and sex-specific risks of myocarditis and pericarditis following Covid-19 messenger RNA Vaccines: a nationwide assessment" by Vu et al., aimed to estimate risks of myocarditis and pericarditis by Covid-19 vaccine type (Pfizer and Moderna) and across sex and age groups, using a matched case-control study design (a separate one for each outcome). Information was collected through linkage of the French National Health Data System with the Covid-19 vaccination database. The results indicated increased risks of both myocarditis (mostly in younger people, and in males and females) and pericarditis (mostly in older subjects) during the first week following both the first and the second vaccine doses, for both vaccine types. Age and sex-specific estimates of excess cases attributable to vaccination revealed a substantial burden of both myocarditis and pericarditis.

GENERAL COMMENTS

The use of administrative data in a medical research present some inherent disadvantages, some of which are addressed by the authors in the 'Discussion' section. However, the results of the

study confirm previous findings and add some new observations, for example regarding perimyocarditis risk in older subjects (less studied so far) as well as myocarditis risks in young females, not only males. COVID-19 mRNA vaccines are still "young", and therefore gathering information about their potential adverse effects is of high importance.

Thank you for your assessment and suggestions.

SPECIFIC COMMENTS

Methods -

Please comment on the completeness of both databases that were used.

Were outcomes included only if occurred up to 21 days following vaccination? that makes sense for the Pfizer vaccine, where the gap between the two doses was usually of 21 days, but for the Moderna vaccine, the gap was at least 28 days, so in theory a later outcome could have also been included. Where there cases of people vaccinated with both vaccine types? how were they treated? or was that an exclusion criterion?

There was no restriction of outcomes based on exposure. All cases of myocarditis and pericarditis were included, irrespective of the time since vaccination, if any.

Yes, there were few heterologous vaccination schemes, in both ways: 2.2% of cases receiving a second dose of mRNA-1273 received a first dose of BNT162b2, and 0.6% the other way around. In the main analysis, we only considered the association with the most recent dose, with no exclusion.

We added this clarification in the methods section.

Discussion -

Please comment on the accuracy of the ICD-10 codes for myocarditis and pericarditis in the database used (reproducibility, validity), since this could be a source for outcome misclassification, which may be differential.

Please refer to the response to comment # 8 of reviewer 2.

Furthermore, we assessed the specificity of the diagnoses by characterizing post-discharge drug treatments that appeared typical of myocarditis and pericarditis.

As for the differential coding error, that would occur in favouring the diagnosis of a known adverse events among exposed. To appreciate this, we ran an analysis for the period when myo/pericarditis were not yet advertised as adverse events of vaccination and found similar associations between vaccination and carditis.

REVIEWERS' COMMENTS

Reviewer #2 (Remarks to the Author):

“Thank you for the author’s efforts in revising the manuscript. They further conducted an interaction analysis between myocarditis history and vaccination status and found no significant interaction. They also appropriately handled those already hospitalized who had little chance of being exposed in a sensitivity analysis. Details of diagnostic procedures by clinicians have been inserted.

In support of the discussion on causality, there is existing research showing a significant decline of myocarditis in adolescents occurrence after an extended dosing interval between the first and second doses is implemented, the authors may like to refer to this (<http://doi.org/10.1001/jamapediatrics.2022.0101>). Apart from this, there are no further comments from me. I congratulate the authors for this excellent work.”

Dear Editor,

Thank you for your help in this final round of revision.

Please find our response in blue arial font, to last reviewer's comment.

Sincerely,

Stéphane Le Vu, on behalf of all co-authors

Reviewer #2 (Remarks to the Author):

“Thank you for the author's efforts in revising the manuscript. They further conducted an interaction analysis between myocarditis history and vaccination status and found no significant interaction. They also appropriately handled those already hospitalized who had little chance of being exposed in a sensitivity analysis. Details of diagnostic procedures by clinicians have been inserted.

In support of the discussion on causality, there is existing research showing a significant decline of myocarditis in adolescents occurrence after an extended dosing interval between the first and second doses is implemented, the authors may like to refer to this (<http://doi.org/10.1001/jamapediatrics.2022.0101>). Apart from this, there are no further comments from me. I congratulate the authors for this excellent work.”

Thank you for your assessment and positive comments.

As for the cited letter, thank you for pointing it out. Although the authors seem to only have compared a single-dose vaccination vs two doses (they just mention the “lengthened interval between doses” as a potential way to reduce myocarditis risk). Still, we included this reference.